# ^18^F-FDG PET-Derived Volume-Based Parameters to Predict Disease-Free Survival in Patients with Grade III Breast Cancer of Different Molecular Subtypes Candidates to Neoadjuvant Chemotherapy

**DOI:** 10.3390/cancers15102715

**Published:** 2023-05-11

**Authors:** Natale Quartuccio, Pierpaolo Alongi, Luca Urso, Naima Ortolan, Francesca Borgia, Mirco Bartolomei, Gaspare Arnone, Laura Evangelista

**Affiliations:** 1Nuclear Medicine Unit, Ospedali Riuniti Villa Sofia-Cervello, 90144 Palermo, Italy; 2Nuclear Medicine Unit, A.R.N.A.S. Ospedali Civico, Di Cristina e Benfratelli, 90127 Palermo, Italy; 3Nuclear Medicine Unit, Oncological Medical and Specialist Department, University Hospital of Ferrara, 44124 Cona, Italy; 4Department of Medicine DIMED, University of Padua, 35128 Padua, Italy

**Keywords:** breast cancer, disease-free survival, semiquantitative parameters, volume-based parameters, ^18^F-FDG, PET/CT

## Abstract

**Simple Summary:**

In breast cancer (BC) patients, neoadjuvant chemotherapy (NAC) is chemotherapy given before surgery. Complete pathological response (pCR) to NAC is defined as the absence of residual tumor cells on microscopy both in the breast and in the axillary or distant lymph nodes in the surgical specimen. Achievement of pCR is associated with longer disease-free survival (DFS), namely survival without tumor recurrence. [^18^F]Fluorodeoxyglucose (^18^F-FDG) positron emission tomography may represent a complementary imaging tool in predicting pCR to NAC. We found (PET)-derived parameters predicting DFS in patients with BC with different molecular subtypes (based on the presence of specific receptors in the tumor). Furthermore, we demonstrated that some of these parameters predicting DFS are significantly different in patients achieving pCR compared to those without pCR after NAC. Larger study samples are needed to confirm these preliminary findings.

**Abstract:**

We investigated whether baseline [^18^F] Fluorodeoxyglucose (^18^F-FDG) positron emission tomography (PET)-derived semiquantitative parameters could predict disease-free survival (DFS) in patients with grade III breast cancer (BC) of different molecular subtypes candidate to neoadjuvant chemotherapy (NAC). For each ^18^F-FDG-PET/CT scan, the following parameters were calculated in the primary tumor (SUVmax, SUVmean, MTV, TLG) and whole-body (WB_SUVmax, WB_MTV, and WB_TLG). Receiver operating characteristic (ROC) analysis was used to determine the capability to predict DFS and find the optimal threshold for each parameter. Ninety-five grade III breast cancer patients with different molecular types were retrieved from the databases of the University Hospital of Padua and the University Hospital of Ferrara (luminal A: 5; luminal B: 34; luminal B-HER2: 22; HER2-enriched: 7; triple-negative: 27). In luminal B patients, WB_MTV (AUC: 0.75; best cut-off: WB_MTV > 195.33; SS: 55.56%, SP: 100%; *p* = 0.002) and WB_TLG (AUC: 0.73; best cut-off: WB_TLG > 1066.21; SS: 55.56%, SP: 100%; *p* = 0.05) were the best predictors of DFS. In luminal B-HER2 patients, WB_SUVmax was the only predictor of DFS (AUC: 0.857; best cut-off: WB_SUVmax > 13.12; SS: 100%; SP: 71.43%; *p* < 0.001). No parameter significantly affected the prediction of DFS in patients with grade III triple-negative BC. Volume-based parameters, extracted from baseline ^18^F-FDG PET, seem promising in predicting recurrence in patients with grade III luminal B and luminal B- HER2 breast cancer undergoing NAC.

## 1. Introduction

Breast cancer (BC) is one of the most frequent cancers in women [1]. The World Health Organization (WHO) cites several variables that may contribute to BC development, including obesity, electromagnetic pollution, environmental pollution, smoking, alcohol intake, and breast implants [2]. In addition, locally advanced breast cancer (LABC) occurs in approximately 30% of patients with breast malignancy at diagnosis [3].

Histological stratification represents the gold standard for the classification of BC, based primarily on the differentiation grade and the expression of estrogen receptor (ER), progesterone receptor (PR), and human epidermal growth factor receptor 2 (HER2) [4]. Several studies demonstrated a significant degree of inter- and intratumoral heterogeneity and distinct BC subtypes, each corresponding to a differentiation state of mammary cells, representing a convergence point between normal mammary gland biology and breast cancer biology. One of the major hypotheses is that breast tumors consist of a mixture of inter-converting breast cancer subtypes that may affect the patient outcome [5]. Nevertheless, grading remains an important prognostic factor for prognosis, with grade III being the more aggressive one [6]; it follows that it is helpful to divide the study samples according to the different grades to make the study population homogenous.

According to clinical guidelines [7], neoadjuvant chemotherapy (NAC), followed by surgery and adjuvant systemic and local treatment, represents the current preferred treatment strategy [3]. Indeed, it can be helpful in early-stage LABC patients to enable breast-conserving surgery (BCS), thus limiting the diffusion of micrometastases and testing potential drug resistance [2,8].

Positron emission tomography (PET), integrated with computed tomography (CT), is a multimodal examination providing volumetric distribution of positron-emitting radionuclides in the human body [9]. [^18^F] Fluorodeoxyglucose (^18^F-FDG) is the most used radiopharmaceutical in oncology and plays a role in the diagnostic workflow of BC, specifically as a metabolic biomarker evaluating glycolytic activity in tumors [10].

In the primary staging of LABC, most updated European Society Medical Oncology guidelines [11] suggest performing ^18^F-FDG PET/CT only in case of inconclusive findings at conventional imaging (CI) or in patients with high-risk disease (i.e., triple-negative breast cancer of aggressive luminal subtypes). Indeed, ^18^F-FDG PET/CT proved to be a valuable tool for detecting axillary lymph node metastases, although sentinel node biopsy still represents the standard gold technique for N-staging [12]. Moreover, its role is recognized for assessing distant metastases, outperforming bone scans and CT for detecting bone metastases [13]. Conversely, the role of ^18^F-FDG PET/CT is limited for evaluating primary BC, except for detecting the primary tumor in CUP syndrome [11]. Nevertheless, the evaluation of chemosensitivity and early response to therapy is clinically relevant in NAC patients, strongly influencing their treatment management and prognosis [14]. In this context, breast magnetic resonance imaging (MRI) is the gold standard for assessing the primary tumor. At the same time, ^18^F-FDG PET/CT can be considered a complementary tool to support MRI in assessing response to therapy [15,16]. Achievement of pathologic complete response (pCR) after NAC is associated with better prognosis in BC patients undergoing surgery, especially when more aggressive subtypes are present [17]. Nevertheless, few articles evaluated the prognostic value of ^18^F-FDG PET/CT before NAC. Instead, they assessed the predictive capability of PET-derived semiquantitative parameters on survival outcomes related to differentiation grade and pathologic characteristics [18,19].

This study’s main objective was to investigate whether ^18^F-FDG PET-derived semiquantitative parameters could predict disease-free survival (DFS) in patients with grade III BC of different molecular types scheduled for NAC. The secondary objective was to evaluate whether the semiquantitative parameters predictive of DFS significantly differed in patients achieving pCR compared to those without pCR after NAC.

## 2. Methods

### 2.1. Patients

Patients were retrospectively retrieved from the databases of two Italian hospitals. Inclusion criteria were: (1) histologically proven diagnosis of grade III BC; (2) history of NAC; (3) execution of a baseline ^18^F-FDG PET/CT scan before the start of NAC; (4) a minimum follow-up time of 3 years from the date of the baseline PET/CT scan to the last available clinical record; (5) curative-intent surgery after NAC.

Patients were classified into five molecular subgroups (Luminal A, Luminal B, Luminal B + HER-2, HER-2 enriched, triple-negative) according to the St. Gallen consensus [20], as already described in a previous paper of our group [21].

For assessing the response to NAC, the Sataloff criteria were considered [22]. In addition, based on the surgical pathology report, a complete pathological response (pCR) was defined as the complete absence of invasive residual tumor cells on microscopy both in the breast and in the axillary or distant lymph nodes.

The DFS was calculated as the time from the baseline ^18^F-FDG PET/CT scan to the first documented disease recurrence or death date. In addition, long-term follow-up data were retrieved from the records of hospital databases, considering a minimum interval of 3 years from the baseline PET/CT scan to the last available clinical record.

Informed consent for inclusion in potential retrospective research studies was signed by every patient in accordance with local authority regulations. In addition, this study was approved by the local institutional review boards with a waiver of authorization due to its retrospective design.

### 2.2. ^18^F-FDG-PET/CT Scanning and Analysis

At both institutions, all patients fasted at least 8 h before PET/CT scan, and after the ^18^F-FDG injection (3.7 MBq/Kg), patients rested on a comfortable chair; the PET/CT scan was performed 60 ± 10 min after the ^18^F-FDG injection following a standard acquisition protocol according to the EANM guidelines [10]. ^18^F-FDG PET/CT scan was performed on both institutions on a dedicated PET/CT system (both scanners were by Siemens) that combines a “full ring” PET scanner and a spiral CT scanner. Emission images ranging from the base of the skull to mid-thigh were acquired for 2–3 min per bed position.

No contrast agent was administered to acquire the CT scan. Data obtained with the CT scan were used for attenuation correction of the PET images. PET images were reconstructed using a dedicated commercial workstation (Syngo. via Workstation, Siemens Healthineers, Enlargen, Germany) along the axial, coronal, and sagittal planes. The Gaussian filter was applied. The data were reconstructed over a 128 × 128 matrix with a 4.75 mm pixel size and 2 mm slice thickness using an ordered-subset expectation maximization algorithm consisting of 4 iterations with 16 subsets.

A volume of interest (VOI) was manually defined by two nuclear medicine physicians on the primary tumor. Further VOIs were placed on each patient in all the suspicious lymph nodes and distant metastases, as described in a previous study by our group [21]. For each ^18^F-FDG-PET/CT scan, the following parameters were calculated in the primary tumor (SUVmax, SUVmean, MTV, TLG). In addition, whole-body semiquantitative parameters (WB_SUVmax, WB_MTV, and WB_TLG) were obtained through the sum of the semiquantitative value of every lesion detected in the scan.

Two independent nuclear medicine physicians with more than five years of experience in imaging reviewed the PET/CT images. In case of discordance, a consensus was reached.

### 2.3. Statistical Analysis

Categorical variables were described as frequencies, whereas continuous variables were presented as mean ± standard deviation or median (range).

Receiver operating characteristic (ROC) analysis was used to determine the optimal threshold for each parameter capability to predict DFS. DFS differences between groups, according to the optimal threshold, were assessed using Kaplan-Meier curves and compared by log-rank test. A *p*-value of 0.05 was used as the minimum threshold for significance. Moreover, the student’s *t* test was used to explore any significant difference between predictors of DFS and pCR in the whole study population. In contrast, non-parametric tests were employed to assess any difference between predictors of DFS and pCR in each molecular subtype. Statistical analysis was performed using MedCalc Statistical Software version 19.1.3 (MedCalc Software, Ostend, Belgium; https://www.medcalc.org; 2020).

## 3. Results

### 3.1. Whole Group Analysis

Ninety-five grade III BC patients with different molecular types were retrieved (luminal A: 5; luminal B: 34; luminal B- HER2: 22; HER2-enriched: 7; triple-negative: 27). Patient characteristics are summarised in Table 1. The median DFS of the whole group was 2162 days (87–3883). There were 29 recurrences at follow-up; detailed information, based on molecular subtypes and location of disease recurrence, is presented in Table 2.

In the group of 95 patients, WB_SUVmax was the best predictor of recurrence (AUC: 0.66; best cut-off: WB_SUVmax > 9.43; SS: 96%; SP: 38.1%; *p* < 0.008). In addition, patients with a WB_SUVmax > 9.43 presented a higher rate of recurrence at the end of follow-up and shorter DFS compared to patients with WB_SUVmax ≤ 9.43 (38.10% vs. 4%; mean DFS: 2552 ± 196 days vs. 3756 ± 124, respectively; *p* = 0.001; Figure 1).

After NAC, 22 patients achieved pCR whereas 61 did not; no information on pCR was available in 12 patients. pCR occurred in 4/27 (15%) patients with luminal B, 6/13 (46%) with luminal B-HER2, 3/6 (50%) with HER2-enriched, and 9/15 (60%) with triple-negative BC. No significant difference was found comparing WB_SUVmax of patients with pCR and patients not achieving pCR after NAC. Limiting the analysis to the 61 patients not achieving pCR, women with WB_SUVmax ≤ 9.43 (n = 20) demonstrated a significantly longer DFS compared to patients with WB_SUVmax > 9.43 (2546 ± 227 days vs. 1769 ± 180 days: *p* = 0.011).

### 3.2. Subgroup Analysis

Patients with luminal A and HER-2-enriched BC subtypes were excluded from the molecular subgroup analysis due to the small samples (5 and 7, respectively). The median DFS for luminal B, B-He and TN tumors were 2284 (177–3883), 1943 (309–3857) and 1405 (87–3778), respectively. Descriptive statistics of all semiquantitative parameters assessed regarding primary tumor, N and M in B, B-He and TN tumors are presented in Table 3.

In grade III luminal B patients (n = 34), WB_MTV (AUC: 0.75; best cut-off: WB_MTV > 195.33; SS: 55.56%, SP: 100%; *p* = 0.002) and WB_TLG (AUC: 0.73; best cut-off: WB_TLG > 1066.21; SS: 55.56%, SP: 100%; *p* = 0.05) were the best predictors of DFS. Patients had a higher probability of DFS in case of a WB MTV ≤ 195.33 (3429 ± 210 vs. 446 ± 144 days; *p* < 0.001; Figure 2) or WB_TLG≤ 1066.33 (937.35 ± 19 vs. 496 ± 131 days; *p* < 0.001; Figure 3). In luminal B patients, there was a trend (*p* = 0.1) for lower median WB_MTV (5.2 vs. 23.18) and WB_TLG (16.93 vs. 137.38) in patients achieving pCR compared to patients not achieving pCR.

In grade III luminal B-HER2 patients (n = 22), WB_SUVmax was the only predictor of DFS (AUC: 0.857; best cut-off: WB_SUVmax > 13.12; SS: 100%; SP: 71.43%; *p* < 0.001); patients with WB_SUVmax ≤ 13.12 had a significantly longer DFS (3857 days) than patients with a WB_SUVmax >13.12 (1693 ± 430 days; *p* < 0.004; Figure 4). No significant difference was found comparing WB_SUVmax of patients with pCR and patients not achieving pCR after NAC.

No parameter significantly affected the prediction of DFS in 27 patients with grade III triple-negative BC.

## 4. Discussion

According to currently available literature, different molecular subtypes of BC are characterized by different FDG-avidity [23,24]. Indeed, a recent meta-analysis analysing 50 studies demonstrated significantly lower SUVmax in luminal A than in luminal B, HE2-positive and TN tumors and a considerably higher SUV max in TN tumors than in luminal B tumors [25].

Nevertheless, regardless of tumor phenotype (luminal, triple negative, or HER2+) and tumor grade, baseline ^18^F-FDG PET/CT appears particularly useful for the initial staging [26] and predicting prognosis [14,18,19,27,28]. Another potential use of ^18^F-FDG PET/CT, according to some groups, is the prediction of response to NAC in patients with BC with luminal-B and Luminal B-HER2 subtypes [21,29].

Luminal-B BC has a higher risk of recurrence and metastasis. Patients with endocrine therapy resistance and chemotherapy insensitivity have a poor prognosis. Indeed, low androgen receptor (AR) expression is associated with poor prognosis in luminal B BC patients. High AR/ER and residual tumor Ki67 were associated with poor DFS in the NAC group. Our study focused on patients with grade III BC, independently from the AR/ER ratio, since these patients have a higher risk of lymph node involvement and recurrence than patients with lower grade [30,31,32].

One of the main issues in BC therapy management regards the possibility of stratifying patients with a significant risk of a progressive disease. In our study, in the whole cohort, without differentiating patients based on molecular type, the only predictor of recurrence was WB_SUVmax (>9.43). Indeed, it was able to stratify the outcomes of both patients with the diverse molecular BC and those with/without pCR at NAC. WB_SUVmax may mirror a more reliable estimate of tumor burden than SUVmax measured in the primary lesion. Indeed WB_SUVmax can provide a more accurate estimate of the tumor burden and glucose metabolism of tumor cells in the whole body and predict prognosis, as demonstrated in previous clinical studies with lymphoma [33] and non-small cell lung cancer [34]. In patients with luminal B-HER2, the cut-off value of WB_SUVmax able to predict a higher risk of recurrence at the 3-year follow-up was superior to 13.12, possibly due to the high rate of osseous, pulmonary, and hepatic metastases occurring in this luminal subtype during follow-up [35]. Furthermore, Kwon and colleagues demonstrated in a large population of BC patients (n = 284), a gradual increase of FDG uptake (measured as SUVmax) from hormonal types to triple negative, also suggesting a trend for higher SUV in HER2-positive tumors compared to tumors without HER2 (*p*  =  0.093) [36].

In luminal B patients, the volume-based parameters, such as WB_MTV and WB_TLG, resulted in the best predictors of DFS, possibly due to the high prognostic power of tumor volume in tumour volume this luminal type and the low FDG uptake compared to other subtypes [37]. Partially in keeping with our results, another study with 143 stage II–III ER+/HER2− BC patients without distant metastases at baseline ^18^F-FDG PET demonstrated the inverse correlation of SUVmax, MTV and TLG with the event-free survival [38]. Volumetric parameters (MTV and TLG) seem more promising as prognostic data than metabolic parameters (SUVmax, SUVpeak, and SUVmean) in patients with grade III Luminal B. Furthermore, WB_TLG and WB_MTV showed a different trend in patients who achieved a pCR after NAC, thus underlining their additional role in predicting the response to therapy before surgery. PET parameters’ different predictive and prognostic roles can be challenging to explain, but future studies could confirm this preliminary data.

As expected, given the intrinsic poor prognosis of patients with grade III triple-negative BC, the study’s results showed the absence of significant parameters able to improve the prediction of DFS in this population. We postulate that the lack of PET-based parameters able to predict DFS in triple-negative BC patients may derive firstly from the study’s main limitation, namely the limited sample of the patient cohort. However, triple-negative is the most aggressive subtype of BC, and any imaging-related outcome predictor would significantly impact patients’ management. Hopefully, radiomics and artificial intelligence could provide added value to answer this unmet clinical need [29], although the lack of standardization is still an obstacle. Secondly, the lack of correlation with other parameters, such as Ki67 values and other biomarkers (e.g., Ca15.3), represents a further limit for more accurate patient stratification. Probably, the combination of more factors could be the best solution. Nevertheless, the statistical significance of the results in this bicentric investigation suggests the possibility of validating our findings in multicentre studies with larger study samples.

Some further limitations should be acknowledged: this is a bi-centre, retrospective study. ^18^F-FDG PET/CT scans were performed on both institutions on a dedicated PET/CT system (both scanners by Siemens Medical Solutions with last-generation technology). Although the PET/CT devices used at the participating centres had not been cross-calibrated, all the acquisitions were executed with the same reconstruction algorithm proposed by the same scanners company, following a standard acquisition protocol defined in the EANM guidelines [10]. These conditions permit optimizing the reproducibility of the semiquantitative parameters analysis. In addition, the number of patients enrolled was inhomogeneous between the two centres due to the different catchment areas. Furthermore, we cannot rule out that the variable treatment regimens performed after NAC during the three years follow-up may have affected the outcome.

## 5. Conclusions

^18^F-FDG PET volume-based parameters demonstrate the potentiality of predicting survival outcomes in patients with grade III luminal B and luminal B-He breast cancer undergoing NAC. Larger study samples are needed to confirm these preliminary findings and their potential application in clinical scenarios.

## Figures and Tables

**Figure 1 cancers-15-02715-f001:**
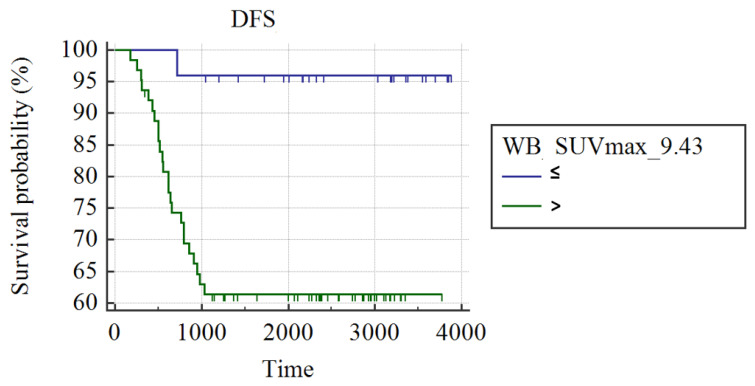
Kaplan Meier curves show different DFS in breast cancer patients with WB_SUVmax ≤ 9.43 and patients with WB_SUVmax > 9.4.3.

**Figure 2 cancers-15-02715-f002:**
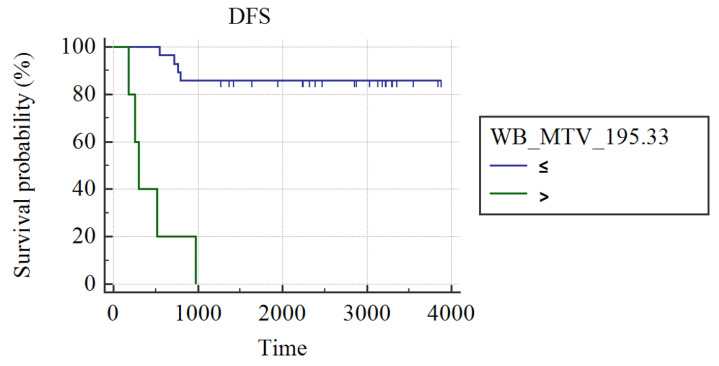
Kaplan Meier curves show different DFS in luminal B patients with WB_MTV ≤ 195.33 and patients with WB_MTV > 195.33.

**Figure 3 cancers-15-02715-f003:**
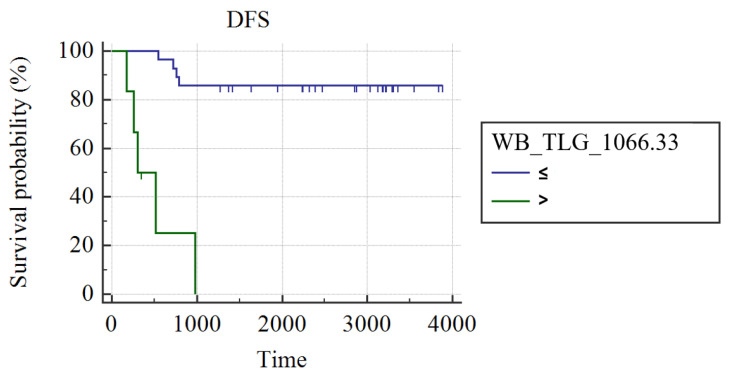
Kaplan Meier curves show different DFS in luminal B patients with WB_TLG ≤ 1066.33 and patients with WB_TLG > 1066.33.

**Figure 4 cancers-15-02715-f004:**
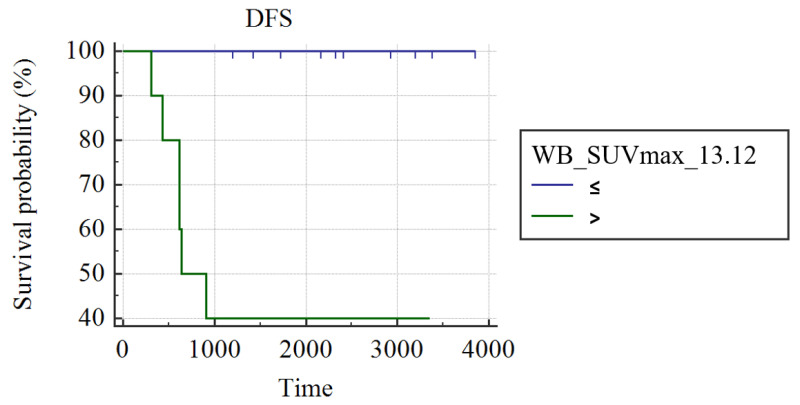
Kaplan Meier curves show different DFS in luminal B-He patients with WB_SUVmax ≤ 13.12 and patients with SUVmax > 13.12.

**Table 1 cancers-15-02715-t001:** Characteristics of the 95 breast cancer patients included in the study.

Molecular subtype	A	5
	B	34
	B-He	22
	Her-2 enriched	7
	Triple-negative	27
Histology	Invasive lobular	8
	Invasive ductal	87
Median age (range) at PET scan, years	50 (26–76)	

**Table 2 cancers-15-02715-t002:** Disease recurrences are based on molecular subtype and location.

	Luminal A	Luminal B	Luminal B-He	Her-2 Enriched	Triple-Negative
Patients with recurrences	0	13	5	1	10
		(38.2%)	(22.7%)	(14.3%)	(37%)
T	0	13	5	1	9
N	0	8	3	1	6
M	0	4	1	1	3

**Table 3 cancers-15-02715-t003:** Summary of semiquantitative parameters for T, N, M and WB, divided for the molecular subtype.

Semiquantitative Parameters		Molecular Subtypes		
		Luminal B	Luminal B-He	Triple Negative
T	SUVmax	11.7 ± 7.5	9.1 ± 5.5	16.5 ± 10.2
	SUVmean	5.5 ± 3.8	4.7 ± 5.5	16.6 ± 10.3
	MTV	70 ± 143.2	5.7 ± 3.9	25.9 ± 46.7
	TLG	340.7 ± 623.4	29 ± 23.6	16.6 ± 10.2
N	SUVmax	10.7 ± 8.4	10.9 ± 8.8	7.1 ± 4.5
	SUVmean	4.9 ± 3.8	5.6 ± 4.2	7.1 ± 4.5
	MTV	22.2 ± 50	39.9 ± 106	6.6 ± 11.5
	TLG	118.3 ± 273	24.3 ± 31.8	27.8 ± 53.9
M	SUVmax	9.9 ± 7.3	6.2 ± 3	5 ± 2.5
	SUVmean	4.6 ± 2.4	3.7 ± 0.9	3.1 ± 1.3
	MTV	4.4 ± 5.3	1.4 ± 1.2	1.6 ± 1.9
	TLG	26.9 ± 38	6.1 ± 6.2	4.9 ± 5
WB	WB_SUVmax	27.5 ± 42.6	63.8 ± 221.8	24.8 ± 15.4
	WB_MTV	88.9 ± 150.1	8.6 ± 5.8	34.7 ± 49.7
	WB_TLG	427.6 ± 684	46.4 ± 42.1	348.8 ± 802

## Data Availability

Data available upon request.

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
