# Peer review of "18F-FDG PET-Derived Volume-Based Parameters to Predict Disease-Free Survival in Patients with Grade III Breast Cancer of Different Molecular Subtypes Candidates to Neoadjuvant Chemotherapy"

_cancers, 2023, doi:10.3390/cancers15102715_

Round 1
Reviewer 1 Report
This bicentric study consist of 95 subjects to investigate retrospectively according to molecular subtypes comparing with FDG-PET volumetric data. Neoadjuvant treatment's success could be incorporated in FDG/PET biological informations. This has a novel objective, probably further studies would strenghten this manuscript.
There are some spelling mistakes.
Author Response
Dear reviewer,
many thanks for appreciating the manuscript.
We revised the manuscript to check and correct misspelling errors.
Reviewer 2 Report
Dear Authors:
The manuscript "18F-FDG PET-derived volume-based parameters to predict disease-free survival in patients with grade III breast cancer of different molecular subtypes candidates to neoadjuvant chemotherapy" by Quartuccio et al has demonstrated that Volume-based parameters, extracted from baseline 18F-FDG PET, seem to be promising tools in the prediction of recurrence in patients with grade III luminal and luminal B- HER2 breast cancer undergoing NAC. I have just a few suggestions.
In introduction, please add more background information about breast cancer (Please cite:
1. Advances in the Prevention and Treatment of Obesity-Driven Effects in Breast Cancers. Front Oncol. 2022 Jun 22;12:820968. doi: 10.3389/fonc.2022.820968. PMID: 35814391; PMCID: PMC9258420.
2. Mitochondrial mutations and mitoepigenetics: Focus on regulation of oxidative stress-induced responses in breast cancers. Semin Cancer Biol. 2022 Aug;83:556-569. doi: 10.1016/j.semcancer.2020.09.012. Epub 2020 Oct 6. Erratum in: Semin Cancer Biol. 2022 Nov;86(Pt 2):1222. PMID: 33035656.)
Best,
Author Response
Dear reviewer, we added the suggested reference and improved the introduction.
Reviewer 3 Report
Dear colleagues,
Thank you for giving me the opportunity to read your manuscript.
I believe the chosen topic is of great clinical relevance and FDG-PET/CT might be there very useful in near future. However, I personally found it very difficult to go through the manuscript without loosing focus on the key messages of your investigations.
A. Introduction:
I would sugges to provide further informations on the role played by FDG-PET/CT at staging (and not recurrence or response assessment which is not the topic of your investigation!) indicating the strengths and challenges (good for M, challenging for N and as stated MRI being the best modality for T).
B. Methods
Your endpoints were not clear to me at this point in the methods section given the formulated goal.
How is DFS defined (this showed up later in the next section, which is I believe confusing)? When assessed? Based on what?
Regarding FDG-PET/CT scans: was contrast medium used? how did you make sure you could compare semiquantative parameters from different scanners? ANOVA-Analysis? Inter-calibration? Did you use the same iterative reconstructions? The assessment of metastases was retrospective based on written report or per blind review? If so, how was the agreement?
Regarding statistical analysis
Please add how continous/categorical variables were described (frequencies vs. mean, etc.)
C. Results
Major concern in my opinion!!!!!!!!!
Absolutely not in accordance with the methods section. A vast majority of the data liste in the method section is missing here.
Also, the presentation of the results doesnt provide a quick and profound understanding of the key messages.
1. No word of recurrence in the methods section, however it appears here in the results section for the first time.
2. Descriptive statistics of recurrence missing; descriptive statistics of all semiquantitative parameters assessed with regards to primary tumor, N and M missing; descriptive statistics of DFS per histological subtype and total missing.
I believe, these major (methodological) weaknesses considerably alter the quality of your presentation and so the understanding of the key messages the autors are trying to explain.
Also further limitations of the investigations were not discussed, such as the used of two different scanners, manual assessment of semiquantitative parameters (at least my understanding), etc...
Author Response
Dear Editor,
We are grateful to the reviewer for the valuable comments.
Please see below our point-by-point responses to all the questions raised by the reviewer.
REVIEWER 3
- Introduction:
I would suggest to provide further information on the role played by FDG-PET/CT at staging (and not recurrence or response assessment which is not the topic of your investigation!) indicating the strengths and challenges (good for M, challenging for N and as stated MRI being the best modality for T).
Author: We agree with the reviewer that this point had to be improved. We tried to rephrase, highlighting the usefulness of FDG PET/CT in N and M primary staging. Moreover, we stated that MRI is the gold standard for T assessment, in particular in the evaluation of response to NAC.
- Methods
Your endpoints were not clear to me at this point in the methods section given the formulated goal.
How is DFS defined (this showed up later in the next section, which is I believe confusing)? When assessed? Based on what?
Author: Dear reviewer, for the sake of clarity we moved and added details to the period regarding the DFS. The period has been moved to the first section “Patients” of the Methods. “The DFS was calculated as the time from the date of the baseline 18F-FDG PET/CT scan (spanning from 2012 to 2019) to the first documented date of disease recurrence or death. Long-term follow-up data were retrieved from the records of hospital databases, considering a minimum interval of 3 years from the baseline PET/CT scan to the last available clinical record.”
Regarding FDG-PET/CT scans: was contrast medium used? how did you make sure you could compare semiquantative parameters from different scanners? ANOVA-Analysis? Inter-calibration? Did you use the same iterative reconstructions? The assessment of metastases was retrospective based on written report or per blind review? If so, how was the agreement?
Author: R1. Only radiopharmaceutical agent was used, no iodine contrast agent was administered. R2. The imaging acquisition protocol, patient preparation and image reconstruction was the same for both the Institutions: all patients were advised to fast for at least 8 h before the integrated PET/CT examination. After injection of about 3 MBq of 18F-FDG/kg/b.w., patients rested for a period of about 60 min in a comfortable chair. Emission images ranging from the base of the skull to mid-thigh were acquired for 2–3 min per bed position. The Gaussian filter was applied to the image after reconstruction along the axial and transaxial directions. The data were reconstructed over a 128 x 128 matrix with 4.75 mm pixel size and 2-mm slice thickness by using an ordered-subset expectation maximization algorithm consisting of 4 iterations with 16 subsets. R3. Two independent nuclear medicine physicians with more than 5 years of experience in imaging reviewed PET/CT images. In case of discordance, a consensus was reached. We added these details in the methods.
Regarding statistical analysis
Please add how continous/categorical variables were described (frequencies vs. mean, etc.)
Author: Categorical variables were described as frequencies, whereas continuous variables were presented as mean ± standard deviation or median. This period was added at the beginning of the session regarding statistical analysis.
- Results
Major concern in my opinion!!!!!!!!!
Absolutely not in accordance with the methods section. A vast majority of the data listed in the method section is missing here. Also, the presentation of the results doesn’t provide a quick and profound understanding of the key messages.
Author: Dear reviewer we extensively modified the results section according to your suggestions.
- No word of recurrence in the methods section, however it appears here in the results section for the first time.
Author: We modified the methods accordingly.
- Descriptive statistics of recurrence missing; descriptive statistics of all semiquantitative parameters assessed with regards to primary tumor, N and M missing; descriptive statistics of DFS per histological subtype and total missing.
Author: R1: We added a table (table 2) to provide an overview of recurrences based on luminal type and anatomical location of recurrence. R2 We added a table (table 3) to provide an overview of the semiquantitative parameters based on the luminal type. R3 We added descriptive statistics of DFS in the results section.
I believe, these major (methodological) weaknesses considerably alter the quality of your presentation and so the understanding of the key messages the authors are trying to explain. Also further limitations of the investigations were not discussed, such as the used of two different scanners, manual assessment of semiquantitative parameters (at least my understanding), etc...
Author: We included the limitations of the study at the end of the discussion section as follow:” Some limitations should be acknowledged: first, this is a bi-centre, retrospective study. 18F-FDG PET/CT scans were performed on both institutions on a dedicated PET/CT system (both scanners by Siemens Medical Solutions with last generation technology). Despite the PET/CT devices used at the participating centres had not been cross-calibrated, all the acquisition were executed with the same reconstruction algorithm, proposed by the same company of the scanners, following a standard acquisition protocol as defined on the EANM guidelines [9]. These conditions permit to optimize the reproducibility of the semiquantitative parameters analysis. In addition, the number of patients enrolled was inhomogeneous between the 2 centres due to the different catchment area. Furthermore, we cannot rule out that the variable treatment regimens performed after NAC during the three years follow-up may have affected the outcome.”
Yours sincerely,
Natale Quartuccio, MD
Round 2
Reviewer 2 Report
strongly suggest for publication.
Reviewer 3 Report
Dear colleagues,
I am thrilled to see that you significantly improved the quality of manuscript following my previous comments.
I believe, the authors addressed my previously listed concerns.
No further requests.
My congratulations!